# Deciphering the Extremes: A Novel Approach for Pathological Long-tailed Recognition in Scientific Discovery

**Zhe Zhao**[1,2][*] **HaiBin Wen**[2][*] **Xianfu Liu**[3] **Rui Mao**[1] **Pengkun Wang**[1][†]
**Liheng Yu**[1] **Linjiang Chen**[1] **Bo An**[4] **Qingfu Zhang**[2] **Yang Wang**[1][†]
[1]University of Science and Technology of China   [2]City University of Hong Kong
[3]China University of Mining and Technology   [4]Nanyang Technological University

## Abstract

Scientific discovery across diverse fields increasingly grapples with datasets exhibiting pathological long-tailed distributions: a few common phenomena overshadow a multitude of rare yet scientifically critical instances. Unlike standard benchmarks, these scientific datasets often feature extreme imbalance coupled with a modest number of classes and limited overall sample volume, rendering existing long-tailed recognition (LTR) techniques ineffective. Such methods, biased by majority classes or prone to overfitting on scarce tail data, frequently fail to identify the very instances—novel materials, rare disease biomarkers, faint astronomical signals—that drive scientific breakthroughs. This paper introduces a novel, end-to-end framework explicitly designed to address pathological long-tailed recognition in scientific contexts. Our approach synergizes a Balanced Supervised Contrastive Learning (B-SCL) mechanism, which enhances the representation of tail classes by dynamically re-weighting their contributions, with a Smooth Objective Regularization (SOR) strategy that manages the inherent tension between tail-class focus and overall classification performance. We introduce and analyze the real-world ZincFluor chemical dataset ($\mathcal{T} = 137.54$) and synthetic benchmarks with controllable extreme imbalances (CIFAR-LT variants). Extensive evaluations demonstrate our method's superior ability to decipher these extremes. Notably, on ZincFluor, our approach achieves a Tail Top-2 accuracy of $66.84\%$, significantly outperforming existing techniques. On CIFAR-10-LT with an imbalance ratio of $1000$ ($\mathcal{T} = 100$), our method achieves a tail-class accuracy of $38.99\%$, substantially leading the next best. These results underscore our framework's potential to unlock novel insights from complex, imbalanced scientific datasets, thereby accelerating discovery. We provide the detailed code in https://github.com/DataLab-atom/PLTR-SD.

## 1 Introduction

Scientific discovery, spanning disciplines from materials science and drug development to astrophysics and genomics, increasingly relies on harnessing vast datasets. However, a pervasive and often underestimated challenge in these domains is the *pathological long-tailed distribution* of data. Unlike common benchmark datasets (e.g., ImageNet-LT [19], Places365-LT [32]), scientific datasets often exhibit extreme imbalances: a few well-understood or easily observable phenomena constitute the majority classes, while a multitude of rare, novel, or hard-to-characterize instances form an extensive tail. More critically, while many existing highly imbalanced benchmarks feature a large number

---

[*]Equal contribution.
[†]Corresponding author.

39th Conference on Neural Information Processing Systems (NeurIPS 2025).

of classes and a relatively substantial total sample size, the pathological long-tailed distributions encountered in scientific exploration are frequently characterized by a comparatively smaller number of classes coupled with a limited overall sample volume. This scarcity of available information for each tail class imposes even more stringent demands on a model's learning capabilities. This is not an artifact but an intrinsic feature of scientific exploration: groundbreaking discoveries often reside in these sparse tail regions, representing new materials with unique properties, biomarkers for rare diseases, or faint astronomical signals indicative of new physical laws. The criticality of accurately identifying and understanding these tail-class instances in scientific domains cannot be overstated.

Standard deep learning models and existing Long-Tailed Recognition (LTR) techniques [31, 29] often falter with such pathological imbalances . Current LTR methods, whether based on re-sampling [4, 8], re-weighting [7, 2], decoupled training [12], or specific loss designs [18, 3], primarily aim to mitigate head-class dominance. However, with extreme scarcity, re-weighting can overfit to noise, re-sampling may lose or redundantly add information, and decoupled training struggles if initial features for tail classes are poorly learned. These shortcomings are drastically amplified at pathological imbalance levels, leading to CATASTROPHIC FAILURES in identifying scientifically paramount tail instances. For example, in our ZincFluor dataset ($T = 137.54$), rare, valuable fluorescent compounds are often missed, hindering discovery. This paper directly confronts pathological long-tailed recognition in

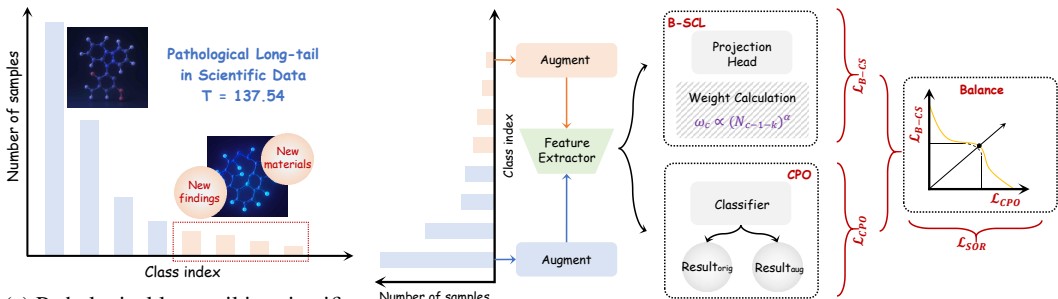

(a) Pathological long-tail in scientific data. Critical findings often reside in rare tail classes.

(b) Our framework: B-SCL ($\mathcal{L}_{\text{B-CS}}$) for tail classes, CPO ($\mathcal{L}_{\text{CPO}}$) for overall accuracy, balanced by SOR ($\mathcal{L}_{\text{SOR}}$).

Figure 1: Visualizing (a) the pathological long-tail challenge in scientific discovery (e.g., $T = 137.54$ in the ZincFluor dataset), where critical findings are in sparse tails, and (b) our proposed framework leveraging Balanced Supervised Contrastive Learning (B-SCL), Classification Performance Objective (CPO), and Smooth Objective Regularization (SOR) to address it.

scientific data. We argue that extreme imbalance necessitates a *paradigm shift* from adapting existing LTR methods to designing bespoke solutions. To this end, we propose a novel, end-to-end trainable framework (overviewed in Figure 1b, with key contributions highlighted below:

▶ **We profoundly unveil and quantify the unique severity of the "pathological long-tail" problem within scientific discovery contexts.** By introducing and analyzing the real-world ZincFluor chemical dataset ($T = 137.54$), and complementing it with synthetic datasets we constructed featuring controllable extreme imbalance (variants of CIFAR-10-LT and CIFAR-100-LT [15]), we systematically benchmark the performance bottlenecks of existing LTR methods in these extreme scenarios, thereby providing new benchmarks and challenges for research in this domain.

▶ **We introduce an innovative balanced supervised contrastive learning framework, inspired by [14], engineered to fundamentally enhance the model's capacity to perceive and represent rare yet critical scientific signals.** Our approach dynamically adjusts the contribution weights of samples from different classes during contrastive learning and integrates multi-objective optimization strategies. This not only compels the model to focus on and learn fine-grained, discriminative features for tail classes but also, through artful loss function design, ensures stable learning of common head-class phenomena. Consequently, it achieves a balanced cognitive understanding across varying class frequencies, effectively preventing the neglect of scarce signals.

▶ **We demonstrate the remarkable efficacy of our method through extensive evaluations.** Critically, on the highly challenging real-world ZincFluor dataset, our approach achieves a breakthrough in identifying rare fluorescent compounds, evidenced by, for instance, a Tail Top-2 accuracy of $66.84\%$, significantly outperforming existing techniques. Furthermore, on synthetic

long-tailed benchmarks with tunable pathological imbalance, our model consistently surpasses state-of-the-art LTR methods, especially when the imbalance is more extreme. For instance, with an imbalance ratio of $1000$ on CIFAR-10-LT ($T = 100$), our method achieves a tail-class accuracy of $38.99\%$, substantially leading the next best method at $28.55\%$. These results underscore the immense potential of our approach to unlock novel insights from complex, imbalanced scientific datasets, offering a potent tool to accelerate scientific discovery.

By developing a robust solution tailored to the pathological long-tailed distributions inherent in scientific research, this work aims to bridge the gap between advanced machine learning capabilities and the pressing need to extract knowledge from the most challenging, yet often most valuable, segments of scientific data.

## 2 Related Work

### 2.1 Long-Tailed Phenomena in Scientific Tasks

Long-tailed distributions, where a few common observations dominate numerous rare ones, are intrinsic to many scientific domains. For instance, in **materials science**, novel materials with exceptional functionalities are far rarer than common stable compounds [1, 20]. Similarly, **drug discovery and genomics** face challenges in identifying rare genetic variants or novel drug targets from vast datasets [5, 26]. **Astrophysics** also encounters this, with rare celestial events or objects being crucial yet sparsely observed compared to common ones [13, 9]. Distinct from typical large-scale LTR benchmarks like ImageNet-LT [19] or Places365-LT [32], scientific datasets often exhibit a *pathological* long-tail: extreme imbalance ratios coupled with a modest number of total classes and often limited overall sample sizes. This unique setting challenges generic LTR methods and motivates our tailored approach.

### 2.2 Long-Tailed Recognition (LTR)

LTR techniques aim to mitigate semantic and structural biases toward majority classes. We categorize prevalent strategies as follows:

- **Data and Loss Manipulation:** Early methods rely on re-sampling (e.g., SMOTE [4] or under-sampling [8]) to balance the training distribution. Re-weighting strategies further refine this by assigning class-specific costs, such as Class-Balanced Loss [7], Focal Loss [18], and LDAM [2]. Notably, recent studies provide a unified theoretical framework for these loss-oriented approaches via localization [28].

- **Decoupled and Manifold Learning:** Decoupled training [12] separates feature learning from classifier adjustment. To enhance the robustness of the learned features, recent works delve into semantic scale imbalance [22] and curvature-balanced feature manifolds [24], aiming for fairer DNNs by optimizing the geometry of perceptual manifolds [23].

- **Logit Adjustment and Distillation:** Post-hoc adjustments, such as label over-smoothing [25] and logit retargeting [21], calibrate the model's confidence. Knowledge distillation [10, 11] and hierarchical label distribution strategies [30] have also proven effective for test-agnostic scenarios.

**Contrastive learning for LTR** has emerged as a potent direction. Building upon Supervised Contrastive Learning (SupCon) [14], methods like Parametric Contrastive Learning (BCL) [6] and Targeted SupCon [17] leverage sample-to-sample relationships to foster discriminative embeddings. Our **B-SCL** extends this paradigm by integrating class-frequency aware weights specifically tailored for *pathological* distributions, where standard benchmarks like iNaturalist [27] fail to capture the extreme scarcity and low total volume typical of scientific discovery datasets.

## 3 Methodology: Balanced Contrastive Representation Learning under Dynamic Multi-Objective Constraints for Pathological Long-Tails

Our methodology addresses the critical challenge of pathological long-tailed recognition, prevalent in scientific discovery, by architecting a synergistic learning framework. This framework prioritizes

the discriminative representation of tail classes while ensuring overall classification efficacy and robustness. We formalize this as a multi-objective optimization problem and derive a tractable loss function that dynamically balances these, often conflicting, objectives.

## 3.1 Formalizing Pathological Long-Tailed Recognition as a Multi-Objective Optimization Problem

We consider a dataset $\mathcal{D} = \{(x_i, y_i)\}_{i=1}^N$ characterized by a pathological long-tailed distribution across $C$ classes, where $x_i \in \mathcal{X}$ and $y_i \in \{0, \ldots, C-1\}$. The per-class sample count $N_c$ exhibits extreme imbalance, quantified by $T = (\max_c N_c)/((\min_c N_c) \cdot C)$. Our goal is to learn model parameters $\theta$ for a feature extractor $f_{\text{backbone}}$, a projection head $\pi_{\text{proj}}$, and a classifier $g_{\text{cls}}$.

In this setting, we identify three primary, potentially conflicting, learning objectives:

1. **Robust Classification Performance ($\mathcal{O}_1(\theta)$):** The model must achieve high classification accuracy across all classes, for both original and augmented data views. This is quantified by the Classification Performance Objective (CPO):

$$\mathcal{L}_{\text{CPO}}(\theta) = \mathbb{E}_{(x,y)\sim\mathcal{D}} \left[ \ell_{\text{CE}}(g_{\text{cls}}(f_{\text{backbone}}(x;\theta)), y) + \ell_{\text{CE}}(g_{\text{cls}}(f_{\text{backbone}}(x';\theta)), y) \right] \tag{1}$$

where $\ell_{\text{CE}}(\mathbf{o}, y) = -\log(\text{softmax}(\mathbf{o})_y)$ is the standard cross-entropy loss. Let $\mathcal{L}_{\text{CE,orig}}(\theta) = \mathbb{E}\left[\ell_{\text{CE}}(g_{\text{cls}}(f_{\text{backbone}}(x;\theta)), y)\right]$ and $\mathcal{L}_{\text{CE,aug}}(\theta) = \mathbb{E}\left[\ell_{\text{CE}}(g_{\text{cls}}(f_{\text{backbone}}(x';\theta)), y)\right]$. Thus, $\mathcal{L}_{\text{CPO}}(\theta) = \mathcal{L}_{\text{CE,orig}}(\theta) + \mathcal{L}_{\text{CE,aug}}(\theta)$.

2. **Tail-Centric Discriminative Representation ($\mathcal{O}_2(\theta)$):** The model must learn highly discriminative features, particularly for information-starved tail classes, to enable their identification. This is addressed by the Balanced Supervised Contrastive Learning (B-SCL) objective:

$$\mathcal{L}_{\text{B-SC}}(\theta) = \lambda_{\text{B-SC}} \cdot \frac{1}{2B} \sum_{\mathbf{z}_j \in \mathcal{S}_{\text{batch}}} w_{y_j} \ell_{\text{SC}}(\mathbf{z}_j; \theta) \tag{2}$$

where $\ell_{\text{SC}}(\mathbf{z}_j; \theta)$ is the standard per-anchor SupCon loss for anchor $\mathbf{z}_j$ with label $y_j$, computed using embeddings $\mathbf{z} = \pi_{\text{proj}}(f_{\text{backbone}}(\cdot; \theta))$. The weights $w_c = \exp(s'_c)/\sum_k \exp(s'_k)$ with $s'_k = (N_{C-1-k})^\alpha$ up-weight tail-class contributions.

The challenge is that minimizing $\mathcal{L}_{\text{CPO}}$ (often dominated by head classes) can conflict with minimizing $\mathcal{L}_{\text{B-SC}}$ (emphasizing tail classes). We seek a solution $\theta^*$ that is Pareto-optimal with respect to $(\mathcal{L}_{\text{CE,orig}}, \mathcal{L}_{\text{CE,aug}}, \mathcal{L}_{\text{B-SC}})$.

**Optimization Target 1 (Constrained Multi-Objective Formulation)** *We aim to find parameters $\theta^*$ that minimize a primary combined objective while ensuring no individual sub-objective becomes excessively large. This can be conceptualized as:*

$$\begin{aligned} \min_\theta \quad & \mathcal{L}_{CPO}(\theta) + \mathcal{L}_{B\text{-}SC}(\theta) \\ \text{subject to} \quad & \mathcal{L}_{CE,orig}(\theta) \leq \epsilon_1 \\ & \mathcal{L}_{CE,aug}(\theta) \leq \epsilon_2 \\ & \mathcal{L}_{B\text{-}SC}(\theta) \leq \epsilon_3 \end{aligned} \tag{3}$$

*where $\epsilon_1, \epsilon_2, \epsilon_3$ are dynamically adjusted upper bounds.*

Solving Optimization Target 1 directly is intractable. Instead, we formulate a penalty-based approach.

## 3.2 Derivation of the Training Objective from Multi-Objective Constraints

To find a solution approximating the Pareto front of $(\mathcal{L}_{\text{CE,orig}}, \mathcal{L}_{\text{CE,aug}}, \mathcal{L}_{\text{B-SC}})$, we employ a scalarization technique that incorporates a penalty for deviations from a balanced state.

**Proposition 1 (LogSumExp as a Smooth Maximum)** *The LogSumExp (LSE) function, $\text{LSE}(\mathbf{v}) = \log \sum_i \exp(v_i)$, is a differentiable, convex approximation of the maximum function, i.e., $\max_i v_i \leq \text{LSE}(\mathbf{v}) \leq \max_i v_i + \log M$ for a vector $\mathbf{v}$ of $M$ components.*

We introduce a Smooth Objective Regularization (SOR) term designed to penalize solutions where any of the fundamental objectives ($\mathcal{L}_{\text{CE,orig}}$, $\mathcal{L}_{\text{CE,aug}}$, or $\mathcal{L}_{\text{B-SC}}$) becomes disproportionately large. This aligns with the Tchebycheff (min-max) approach for multi-objective optimization. Let $\mathcal{L}_{\text{constituent}}(\theta) = [\mathcal{L}_{\text{CE,orig}}(\theta), \mathcal{L}_{\text{CE,aug}}(\theta), \mathcal{L}_{\text{B-SC}}(\theta)]^T$. The SOR term is defined as:

$$\mathcal{L}_{\text{SOR}}(\theta) = \lambda_{\text{SOR}} \cdot \text{LSE}(\mathcal{L}_{\text{constituent}}(\theta)/\tau_{\text{SOR}}) \tag{4}$$

where $\lambda_{\text{SOR}}$ is a regularization strength and $\tau_{\text{SOR}}$ is a temperature parameter. For simplicity and alignment with the paper's practical implementation, we set $\tau_{\text{SOR}} = 1$. Thus,

$$\mathcal{L}_{\text{SOR}}(\theta) = \lambda_{\text{SOR}} \cdot \log\left(\exp(\mathcal{L}_{\text{CE,orig}}(\theta)) + \exp(\mathcal{L}_{\text{CE,aug}}(\theta)) + \exp(\mathcal{L}_{\text{B-SC}}(\theta))\right). \tag{5}$$

The final training objective $\mathcal{L}_{\text{total}}(\theta)$ combines the primary objectives with this dynamic regularization:

$$\mathcal{L}_{\text{total}}(\theta) = \underbrace{\mathcal{L}_{\text{CE,orig}}(\theta) + \mathcal{L}_{\text{CE,aug}}(\theta)}_{\mathcal{L}_{\text{CPO}}(\theta)} + \mathcal{L}_{\text{B-SC}}(\theta) + \mathcal{L}_{\text{SOR}}(\theta). \tag{6}$$

Substituting Eq. 5 into Eq. 6:

$$\begin{aligned} \mathcal{L}_{\text{total}}(\theta) = \mathcal{L}_{\text{CPO}}(\theta) + \mathcal{L}_{\text{B-SC}}(\theta) \\ + \lambda_{\text{SOR}} \cdot \log\left(\exp(\mathcal{L}_{\text{CE,orig}}(\theta)) + \exp(\mathcal{L}_{\text{CE,aug}}(\theta)) + \exp(\mathcal{L}_{\text{B-SC}}(\theta))\right). \end{aligned} \tag{7}$$

**Theoretical Justification.** Minimizing $\mathcal{L}_{\text{total}}(\theta)$ aims to achieve a state where: 1. The sum of the primary objectives ($\mathcal{L}_{\text{CPO}} + \mathcal{L}_{\text{B-SC}}$) is low. 2. The SOR term, leveraging Proposition 1, ensures that the maximum of the constituent objectives ($\mathcal{L}_{\text{CE,orig}}, \mathcal{L}_{\text{CE,aug}}, \mathcal{L}_{\text{B-SC}}$) is also kept low.

This formulation implicitly seeks a solution where no single objective can be significantly improved without degrading another, which is characteristic of Pareto-optimal solutions. The SOR term dynamically adjusts the pressure on each constituent objective. If, for instance, $\mathcal{L}_{\text{B-SC}}$ becomes very large (e.g., due to difficulty in representing extremely rare tail classes or overfitting), the gradient contribution from the SOR term with respect to $\mathcal{L}_{\text{B-SC}}$ will increase, effectively pushing the optimizer to reduce it. Similarly, if $\mathcal{L}_{\text{CE,orig}}$ is high (poor classification on original data), SOR will penalize this.

This dynamic balancing is crucial for pathological long-tails:

- **B-SCL ($\mathcal{O}_2$)** provides the necessary focus on tail classes by up-weighting their contribution to representation learning, fostering discriminative features despite data scarcity.
- **CPO ($\mathcal{O}_1$)** ensures general classification utility.
- **SOR** acts as the arbiter, preventing either the tail-class specific learning or the general classification learning from excessively dominating and destabilizing the other, thus guiding the optimization towards a robust equilibrium suitable for the extreme imbalances encountered in scientific discovery. The **Appendix C** provides more theory.

## 4  Experiments

In this section, we conduct extensive experiments to evaluate the efficacy of our proposed method, referred to as **Ours**, in addressing pathological long-tailed recognition. We first detail the datasets and evaluation metrics (Section 4.1). We then outline the experimental setup, including baselines and implementation details (Section 4.2). Subsequently, we present quantitative results on both real-world scientific datasets and synthetic long-tailed benchmarks (Section 4.3), followed by ablation studies (Section 4.4) and qualitative analyses (Section 4.5).

### 4.1  Datasets, Metrics, and Pathological Imbalance

The variable $\mathcal{T}$ is used to quantify the degree of pathological imbalance in the dataset. A higher value of $\mathcal{T}$ corresponds to a more pronounced imbalance. It is defined as:

$$\mathcal{T} = \frac{N_{\text{majority}}}{N_{\text{minority}} \cdot N_{\text{classes}}} \tag{8}$$

where $N_{\text{majority}}$ represents the number of samples in the majority class, $N_{\text{minority}}$ represents the number of samples in the minority class, and $N_{\text{classes}}$ denotes the total number of classes.

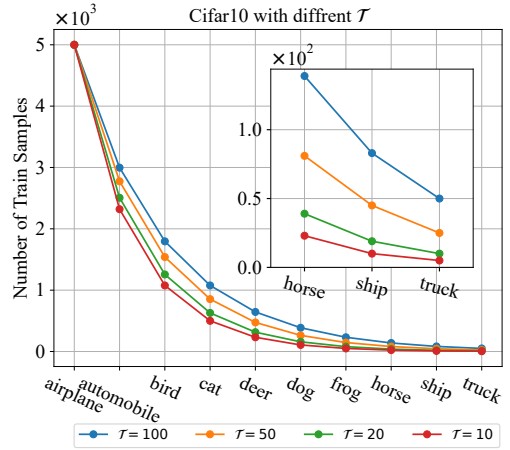 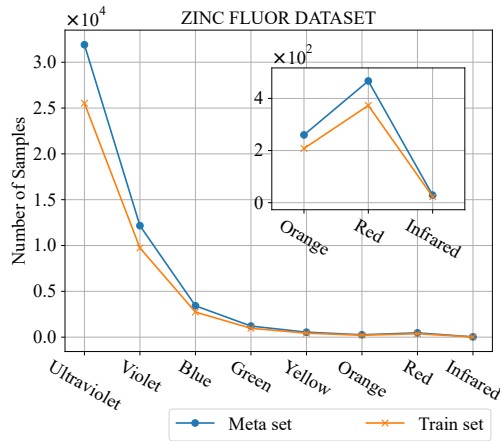

(a) Training sample distribution per class in **CIFAR-10-LT** under different $\mathcal{T}$ settings.

(b) Comparison of meta-samples per class with training samples in **ZincFluor**.

Figure 2: Dataset characteristics: (a) **CIFAR-10-LT** class distributions. (b) **ZincFluor** sample counts.

Table 1: The anonymized **ZincFluor** dataset examples.

| Index | SMILES | Pred Fluor Colour | Intensity | Fluor Value |
|-------|--------|-------------------|-----------|-------------|
| ZINC1 | CC(=O)Nc1c(-c2cccccc2)c(C)nn1-c1ccc(C(=O)Nc2ccc... | Ultraviolet | Weak | 1 |
| ZINC2 | Cc1nc(-c2cccc(NC(=O)c3ncccn3)c2)cs1 | Ultraviolet | Weak | 1 |
| ZINC3 | CCCc1ccc(/N=N/C(Sc2nnc(-c3ccncc3)o2)=C(O)c2ccc... | Ultraviolet | Weak | 1 |
| ZINC4 | CCOC(=O)Nc1ccc2c(Sc3ccccc[n+]3[O-])cc(=O)oc2c1 | Violet | Weak | 2 |
| ZINC5 | O=CNC(=O)c1sc2ncccc3c2c1ncn3-c1cccccc1 | Violet | Weak | 2 |
| ZINC6 | Cc1ccn(C(=O)c2cccc(N3CCCS3(=O)=O)c2)c=NC2CCCC... | Blue | Weak | 3 |

**Real Dataset: ZincFluor.** This is a classification dataset from a chemical laboratory. Its general content is exemplified in Table 1. As shown in Figure 2b, the dataset exhibits an extremely pathological class imbalance with an imbalance degree $\mathcal{T} = 137.54$ after an 8:2 train-test split. This severe imbalance poses a significant challenge to existing long-tailed learning methods. The dataset comprises 8 distinct fluorescence levels used as classes.

**Synthetic Datasets: CIFAR-LT.** To comprehensively evaluate robustness, we use long-tailed variants of **CIFAR-10** and **CIFAR-100** [15] (i.e., **CIFAR-10-LT** and **CIFAR-100-LT**). We control the imbalance ratio (IR = $N_{\text{majority}}/N_{\text{minority}}$) to construct datasets with varying degrees of pathological imbalance $\mathcal{T}$. Figure 2a visualizes the training sample distribution across classes in **CIFAR-10-LT** under different $\mathcal{T}$ settings.

**Evaluation Metrics.** We report Top-1 accuracy as the primary metric. For **ZincFluor**, we show per-class Top-1 accuracy and aggregated tail-class accuracies (Tail Top-6, Top-4, Top-2). For **CIFAR-LT**, we report overall Top-1 accuracy ("All"), and accuracies on "Head", "Medium", and "Tail" class splits based on training sample counts.

## 4.2 Experimental Setup

**Baselines.** To rigorously evaluate our framework, we compare **Ours** against a diverse set of representative LTR baselines: (1) **CE**: Standard Cross-Entropy training. (2) **BS** [19]: Balanced Softmax for logit adjustment. (3) **BCL** [6]: Balanced Contrastive Learning. (4) **CE-DRW** and **LDAM-DRW** [2]: Re-weighting and margin-based loss combined with Deferred Re-Weighting. (5) **KPS** [16]: Key Point Sensitive loss focusing on tail-class anchors. (6) **LORT** [21]: Logits Retargeting approach. For the ablation study on **ZincFluor**, the "base" refers to the LOS-based method (Logits Over-Smoothing) [25], which serves as our primary classification backbone.

**Implementation Details.** All models were implemented using PyTorch and PyTorch Geometric. The experiments were conducted on a single NVIDIA Tesla A100 GPU, with results reported accordingly. Specifically, for the ZincFluor dataset, RDKit was utilized to convert SMILES strings into graph data,

and a backbone network consisting of six stacked GCN layers was employed. During training, the number of epochs for the ZincFluor dataset was set to 100. For all other experiments, configurations followed those of LOS. Models were trained for 200 epochs using the SGD optimizer (learning rate lr=0.01, momentum=0.9, weight decay=5e-3) in conjunction with the CosineAnnealingLR learning rate scheduler.

## 4.3 Quantitative Results

Table 2: Top-1 accuracy on ZincFluor $\mathcal{T} = 137.54$. The grayed-out section indicates the primary observation indicator. **Blod** indicates the best performance while underline indicates the second best.

| Method | Fluor Leval | | | | | | | | Tail Top acc | | |
|---|---|---|---|---|---|---|---|---|---|---|---|
| | 1 | 2 | 3 | 4 | 5 | 6 | 7 | 8 | Top-6 | Top-4 | Top-2 |
| CE | 85.19 | 70.49 | 19.71 | 25.62 | 0.00 | 0.00 | 73.40 | 0.00 | 19.78 | 18.35 | 36.70 |
| BS | 82.73 | 30.66 | 43.21 | 28.51 | 0.00 | 25.00 | 72.34 | 0.00 | 28.17 | 24.33 | 36.17 |
| BCL | 86.45 | 51.17 | 51.82 | 22.31 | 17.43 | 40.38 | 69.15 | 50.00 | 41.84 | 44.24 | 59.57 |
| CE-DRW | 94.52 | 45.62 | 27.59 | 26.86 | 12.84 | 42.31 | 67.02 | 33.33 | 34.99 | 38.87 | 50.17 |
| LDAM-DRW | 91.93 | 47.27 | 28.91 | 20.66 | 22.94 | 28.85 | 69.15 | 33.33 | 33.97 | 38.56 | 51.24 |
| KPS | 91.10 | 45.70 | 51.09 | 23.97 | 1.83 | 19.23 | 71.28 | 0.00 | 27.90 | 23.08 | 35.64 |
| LORT | 72.23 | 25.81 | 1.75 | 33.88 | 0.00 | 26.92 | 75.53 | 0.00 | 23.01 | 25.61 | 37.76 |
| Ours | 90.97 | 42.21 | 58.10 | 21.49 | 11.01 | 34.62 | 67.02 | 66.67 | **43.15** | **44.83** | **66.84** |

**Performance on ZincFluor.** Table 2 details the Top-1 accuracy on **ZincFluor** ($\mathcal{T} = 137.54$). Our method demonstrates highly competitive performance on individual "Fluor Levels" and substantially outperforms all baselines in tail-class focused metrics. Notably, **Ours** achieves a Tail Top-2 accuracy of **66.84**%, a significant improvement over the second-best, **BCL** (59.57%). This underscores our method's capability in handling real-world, pathologically imbalanced scientific data.

Table 3: Top-1 accuracy on CIFAR10-LT with different Imbalance ratio. The grayed-out section indicates the primary observation indicator. **Blod** indicates the best performance while underline indicates the second best.

| Method | IR=1000 $\mathcal{T} = 100$ | | | | IR=500 $\mathcal{T} = 50$ | | | | IR=200 $\mathcal{T} = 20$ | | | | IR=100 $\mathcal{T} = 10$ | | | |
|---|---|---|---|---|---|---|---|---|---|---|---|---|---|---|---|---|
| | Head | Medium | Tail | All | Head | Medium | Tail | All | Head | Medium | Tail | All | Head | Medium | Tail | All |
| CE | 79.03 | 45.90 | - | 56.6 | 81.32 | 53.55 | 7.8 | 61.06 | 81.91 | 47.8 | - | 71.68 | 83.54 | 58.5 | - | 78.53 |
| BS | 76.68 | 64.0 | 16.85 | 62.18 | 76.98 | 69.10 | 30.5 | 66.11 | 82.21 | 61.53 | - | 76.01 | 84.81 | 64.8 | - | 80.81 |
| BCL | 79.82 | 57.3 | 28.55 | 65.06 | 82.22 | 60.05 | 41.25 | 70.79 | 82.47 | 71.50 | - | 79.18 | 83.25 | 81.2 | - | 82.84 |
| CE-DRW | 77.97 | 55.15 | 4.15 | 58.64 | 81.58 | 56.15 | 31.2 | 66.42 | 79.34 | 65.17 | - | 75.09 | 81.94 | 68.9 | - | 79.33 |
| LDAM-DRW | 75.57 | 52.0 | 15.25 | 61.19 | 78.27 | 59.75 | 40.7 | 67.05 | 78.79 | 63.7 | - | 74.29 | 81.98 | 68.55 | - | 79.29 |
| KPS | 78.9 | 56.85 | 6.65 | 60.04 | 78.95 | 45.2 | 42.75 | 64.96 | 82.27 | 57.23 | - | 74.76 | 82.73 | 61.0 | - | 78.38 |
| LORT | 80.75 | 65.30 | 0.05 | 61.52 | 81.0 | 60.0 | 0.05 | 60.61 | 83.36 | 58.50 | - | 75.9 | 83.76 | 85.1 | - | 84.03 |
| Ours | 76.80 | 76.60 | 38.99 | **69.20** | 81.68 | 79.64 | 59.39 | **77.94** | 84.05 | 84.33 | - | **84.14** | 87.59 | 89.80 | - | **88.04** |

Table 4: Top-1 accuracy on CIFAR100-LT with different Imbalance ratio. The grayed-out section indicates the primary observation indicator. **Blod** indicates the best performance while underline indicates the second best.

| Method | IR=500 $\mathcal{T} = 5$ | | | | IR=200 $\mathcal{T} = 2$ | | | | IR=100 $\mathcal{T} = 1$ | | | |
|---|---|---|---|---|---|---|---|---|---|---|---|---|
| | Head | Medium | Tail | All | Head | Medium | Tail | All | Head | Medium | Tail | All |
| CE | 80.96 | 46.15 | 7.37 | 36.59 | 79.07 | 51.55 | 6.87 | 42.38 | 78.09 | 48.51 | 10.97 | 47.6 |
| BS | 78.81 | 50.35 | 14.56 | 40.57 | 74.73 | 55.06 | 18.92 | 46.87 | 75.46 | 52.06 | 27.23 | 52.8 |
| BCL | 78.31 | 51.31 | 14.96 | 40.88 | 76.73 | 53.48 | 20.44 | 47.57 | 74.57 | 52.66 | 26.23 | 52.4 |
| CE-DRW | 77.58 | 47.08 | 13.58 | 38.93 | 74.87 | 52.55 | 18.71 | 46.05 | 75.89 | 51.69 | 22.07 | 51.27 |
| LDAM-DRW | 74.73 | 49.58 | 15.83 | 39.92 | 73.97 | 52.29 | 18.21 | 45.5 | 72.74 | 51.09 | 21.80 | 49.88 |
| KPS | 78.96 | 48.35 | 12.94 | 39.31 | 77.27 | 52.84 | 16.97 | 46.18 | 76.54 | 45.6 | 22.6 | 50.93 |
| LORT | 67.69 | 39.46 | 7.44 | 31.43 | 71.63 | 56.9 | 20.21 | 47.01 | 70.11 | 55.37 | 33.33 | 53.92 |
| Ours | 68.57 | 56.65 | 22.52 | **43.37** | 68.26 | 60.38 | 30.30 | **51.02** | 71.57 | 62.02 | 32.03 | **56.37** |

**Performance on CIFAR-LT Benchmarks.** Across CIFAR-LT benchmarks (Tables 3 4), our method consistently achieves superior overall accuracy and, more critically, demonstrates substantial gains in tail class accuracy across all tested imbalance ratios. For instance, on CIFAR-10-LT with extreme imbalance (IR=1000), our tail accuracy reaches **38.99**%, significantly outperforming BCL (28.55%), alongside leading overall accuracy (**69.20**% vs. 65.06%). This superior tail performance extends to CIFAR-100-LT, where at IR=100, our **32.03**% tail accuracy notably exceeds competitors (e.g., BS 27.23%), and at IR=500, we achieve **22.52**% against BCL's 14.96%, while consistently maintaining

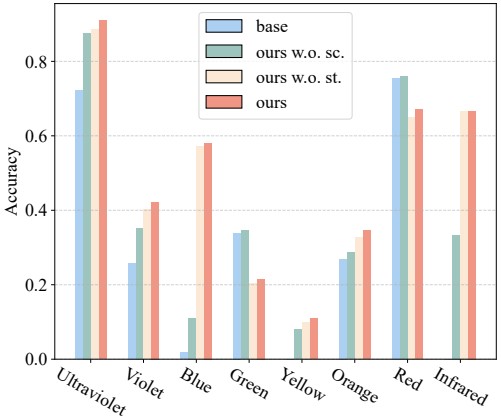 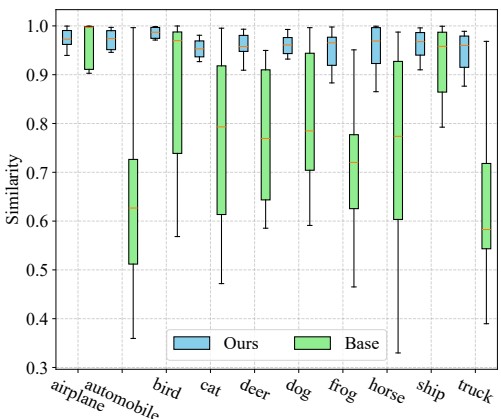

(a) Ablation study on **ZincFluor**. "sc." denotes B-SCL, "st." denotes SOR. Our full method outperforms ablated versions and the base.

(b) Cosine similarity between original and augmented sample features on **CIFAR-10-LT** (IR=10, trained on IR=1000). Our method shows higher robustness.

Figure 3: Ablation study and representation robustness: (a) Component analysis of our method. (b) Feature similarity across augmentations.

the highest overall accuracies. These comprehensive results validate our approach's robustness and effectiveness in enhancing recognition of underrepresented tail classes, particularly under severe imbalance conditions.

## 4.4 Ablation Studies

To dissect the contributions of the core components of our method, we conduct ablation studies on the **ZincFluor** dataset, with results shown in Figure 3a. Removing the Balanced Supervised Contrastive learning loss ("sc.") from our full model ("ours") leads to a significant drop in per-class performance, particularly for the tail classes, highlighting the importance of B-SCL for learning discriminative representations under severe imbalance. Similarly, removing the Smooth Objective Regularization term ("st.") also results in degraded performance compared to the full model, indicating that SOR plays a vital role in balancing the different learning objectives and stabilizing training. The performance of our ablated models still generally surpasses the "base" LOS-based baseline. These studies confirm that both B-SCL and SOR are crucial for achieving the superior performance of our proposed framework.

## 4.5 Qualitative Analysis

**Representation Robustness to Augmentation.** Figure 3b shows the cosine similarity between the model outputs (features) of original samples and their augmented counterparts on **CIFAR-10-LT** (IR=10, models trained on IR=1000). **Ours** generally maintains higher similarity across classes compared to a **Base** method, suggesting that our approach learns representations that are more invariant and robust to data augmentations.

**Class-Level Feature Discriminability.** The quality of learned feature representations is further assessed by visualizing class-level cosine similarity matrices on **CIFAR-10-LT** (IR=1000), as shown in Figure 4. Panel (a) (standard **CE** loss) exhibits a diffuse similarity matrix with poor separation between classes. In contrast, panel (b) (**Ours**) displays a much clearer block-diagonal structure, indicating strong intra-class compactness and high inter-class separability. This demonstrates the superior ability of our method to learn discriminative features, which is fundamental for effective long-tailed recognition.

## 4.6 Discussion of Experimental Findings

The comprehensive experimental results consistently validate the efficacy of our proposed method. The substantial gains observed on the pathologically imbalanced **ZincFluor** dataset, especially in

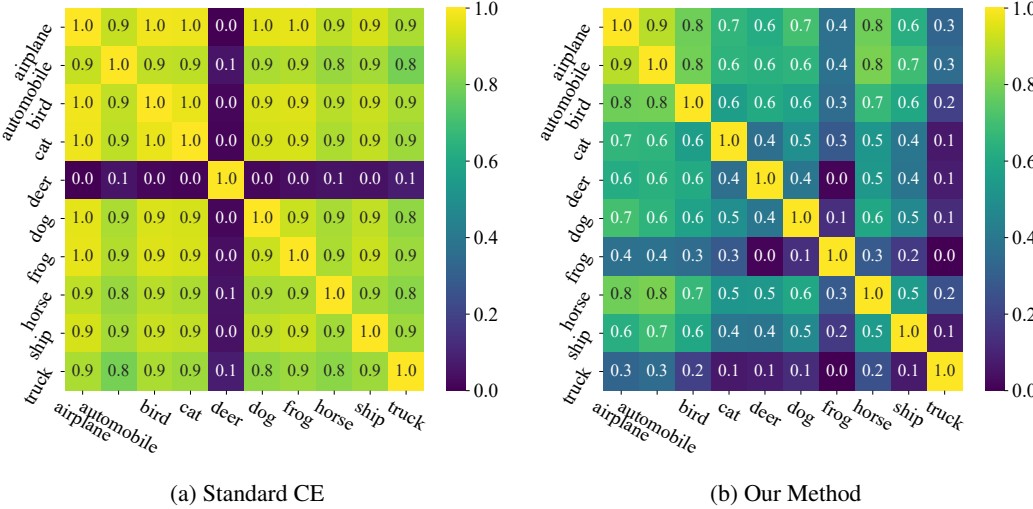

| (a) Standard CE | (b) Our Method |

Figure 4: Class-level feature representation cosine similarities on **CIFAR-10-LT** (IR=1000). (a) Standard cross-entropy loss. (b) Our proposed method, showing improved class separability.

recognizing rare tail classes, highlight its practical utility for scientific discovery tasks. Furthermore, its robust and superior performance across a wide spectrum of imbalance ratios on synthetic **CIFAR-LT** benchmarks underscores its generalizability and strength in handling varying degrees of data imbalance. The ablation studies confirm the synergistic contributions of the B-SCL and SOR components, and qualitative analyses provide visual evidence of the improved representation quality and feature discriminability achieved by our approach. These findings strongly support our central claim that a tailored framework integrating balanced contrastive representation learning with dynamic multi-objective optimization is pivotal for effectively addressing pathological long-tailed recognition.

## 5 Conclusion

This paper tackled the critical issue of pathological long-tailed recognition in scientific discovery, where rare instances crucial for breakthroughs are often missed by standard methods. We introduced a novel framework combining Balanced Supervised Contrastive Learning (B-SCL) to enhance tail-class representation and Smooth Objective Regularization (SOR) to dynamically balance competing learning objectives. Our approach ensures focused learning on sparse tail data without compromising overall performance. Extensive experiments on the real-world ZincFluor dataset and synthetic CIFAR-LT benchmarks with extreme imbalances demonstrated significant improvements over state-of-the-art LTR techniques, particularly in identifying critical tail classes. This work provides a robust tool for extracting valuable insights from severely imbalanced scientific datasets, paving the way for accelerated discovery. Future directions include incorporating domain knowledge and extending to other scientific data modalities.

## Acknowledgements

The authors gratefully acknowledge the support from the National Natural Science Foundation of China (NSFC) under Grant Nos. 62402472, and 12227901. This work was also supported by the Natural Science Foundation of Jiangsu Province (No. BK20240461), the Key Basic Research Foundation of Shenzhen (No. JCYJ20220818100005011), the Research Grants Council of the Hong Kong Special Administrative Region (GRF Project No. CityU 11215723), the Project of Stable Support for Youth Team in Basic Research Field, CAS (No. YSBR-005), and the Academic Leaders Cultivation Program at USTC. The AI-driven experiments, simulations and model training were performed on the robotic AI-Scientist platform of Chinese Academy of Sciences.

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

# A    ZincFluor Dataset: Molecular Fluorescence and Pathological Imbalance

This section provides a more detailed introduction to the ZincFluor dataset used in our experiments (Section 4.2). We describe its origin, the underlying scientific task of molecular fluorescence prediction, and elaborate on why its inherent data distribution constitutes a "pathological long-tail" challenge, distinct from standard long-tailed benchmarks.

## A.1    Molecular Fluorescence and Its Scientific Significance

Molecular fluorescence is a photophysical process where a molecule absorbs light at a specific wavelength and then re-emits light at a longer wavelength. This phenomenon is of immense scientific and technological importance, underpinning applications in diverse fields such as:

- **Materials Science:** Development of organic light-emitting diodes (OLEDs), fluorescent probes for material characterization, and luminescent sensors.
- **Biology and Medicine:** Bioimaging (using fluorescent tags or probes), drug discovery (screening for compounds with desired fluorescence properties or using fluorescent markers), and diagnostic assays.
- **Chemistry:** Understanding molecular electronic structure, reaction monitoring, and analytical techniques.

Discovering and synthesizing novel molecules with tailor-made fluorescence properties (e.g., specific emission wavelengths, high quantum yield, photostability, environmental sensitivity) is a core pursuit in chemistry and materials science. Predicting these properties computationally from molecular structure is a valuable tool to accelerate this discovery process, avoiding costly and time-consuming experimental synthesis and screening.

## A.2    The ZincFluor Dataset: Structure and Task

The ZincFluor dataset originates from experimental measurements conducted in a chemical laboratory. It comprises a collection of molecular compounds, each represented by its SMILES (Simplified Molecular Input Line Entry System) string, which provides a concise textual description of the molecule's structure. For each compound, an associated fluorescence property has been measured and categorized into one of 8 distinct levels. These levels, serving as the classification labels in our task, represent different aspects of the molecule's fluorescent behavior, such as intensity or emission characteristics (exemplified by the 'Pred Fluor Colour', 'Intensity', and 'Fluor Value' columns in Table 1 in the main paper). The task is to predict the fluorescence level of a molecule given its SMILES string (or its graph representation derived from it, as described in Section **??**).

## A.3    Pathological Imbalance in ZincFluor

As highlighted in the main paper (Figure 2b), the ZincFluor dataset exhibits an exceptionally severe class imbalance across its 8 fluorescence levels. Following an 8:2 training-test split, the imbalance factor $T$ (defined as $N_{\text{majority}}/(N_{\text{minority}} \cdot N_{\text{classes}})$) is calculated to be $137.54$. This metric quantifies the extreme disparity between the most frequent and least frequent fluorescence levels relative to the total number of classes.

The distribution is not merely skewed; it is **pathologically** imbalanced in the context of scientific discovery for the following key reasons:

1. **Scientific Value Concentrated in the Tail:** The rare fluorescence levels (minority classes) often correspond to molecules exhibiting unusual, extreme, or highly specific fluorescent properties. These are precisely the properties that are most sought after for cutting-edge applications (e.g., a molecule with exceptionally high brightness for bioimaging, or a compound emitting light at a very specific wavelength for sensing). In essence, the "rare" instances in this dataset represent the potential scientific breakthroughs or novel materials, not just less common variations of typical compounds.

2. **Limited Sample Volume:** Unlike large-scale public benchmarks like ImageNet-LT or Places365-LT which have millions of images and hundreds/thousands of classes, scientific

datasets like ZincFluor often arise from costly and labor-intensive experimental processes. The combination of extreme imbalance and limited total data volume means that the tail classes have critically few samples, sometimes only a handful, making robust learning for these classes incredibly difficult.

3. **Fundamental Task Objective:** The implicit goal in analyzing such scientific data is often to discover or identify these rare, high-value instances for further investigation. A model that performs well on the frequent (head) classes but fails to identify the rare fluorescent compounds in the tail effectively misses the primary scientific objective.

The ZincFluor dataset represents a "pathological long-tail" problem because the tail instances are not just rare data points, but are scientifically **critical** signals buried within a sea of common observations. Standard LTR methods, primarily designed to mitigate majority bias in general classification tasks, struggle significantly with this unique combination of extreme imbalance, limited data, and the paramount importance of correctly identifying the scarce tail instances that drive scientific advancement. Our work is specifically motivated to address this particular challenge.

# B   Evaluation on Custom Places-LT Datasets with Reduced Class Counts

To further demonstrate the robustness and generalizability of our method across different domains and varying degrees of pathological imbalance complexity, we conducted additional experiments on modified versions of the standard Places-LT dataset [32]. While the original Places-LT features a large number of classes (365), our definition of pathological long-tail in scientific discovery contexts highlights scenarios with extreme imbalance coupled with a modest number of classes and limited overall sample volume(Section 1). To better align with this, we constructed synthetic long-tailed datasets derived from Places-LT that maintain a high imbalance ratio but reduce the total number of classes.

## B.1   Custom Places-LT Dataset Construction

We generated new synthetic datasets based on the original Places-LT dataset (with an imbalance ratio of 996) by performing class-level sampling. Specifically, we followed a procedure to select a subset of classes:

- We identified the class with the maximum number of samples (most frequent) and the class with the minimum number of samples (least frequent) in the original Places-LT dataset. These two classes were always included in our custom datasets.

- From the remaining classes in the original Places-LT, we uniformly sampled additional classes until the desired total number of categories was reached.

Using this procedure, we constructed three new datasets containing a total of 10, 50, and 100 classes, respectively. These datasets are referred to as "Places-LT (IR 996, 10 Categories)", "Places-LT (IR 996, 50 Categories)", and "Places-LT (IR 996, 100 Categories)".

Importantly, these newly constructed datasets maintain the same imbalance ratio (IR = $N_{\text{majority}}/N_{\text{minority}} = 996$) as the original Places-LT. However, by significantly reducing the total number of classes while keeping a very high IR, these datasets exhibit an even more pronounced "pathological" nature in terms of the *relative sparsity of tail classes within a smaller overall class space*, aligning more closely with the characteristics observed in datasets like ZincFluor compared to the original 365-class Places-LT. Figure 5 shows the sample distribution characteristics for these three custom datasets, illustrating the persistent long-tail across varying numbers of categories.

## B.2   Experimental Setup

## B.3   Results and Analysis

Table 5 presents the quantitative results on the three custom Places-LT datasets.

The results show that our method consistently achieves the best overall Top-1 accuracy across all three variants of the custom Places-LT datasets, ranging from 10 to 100 categories, while maintaining

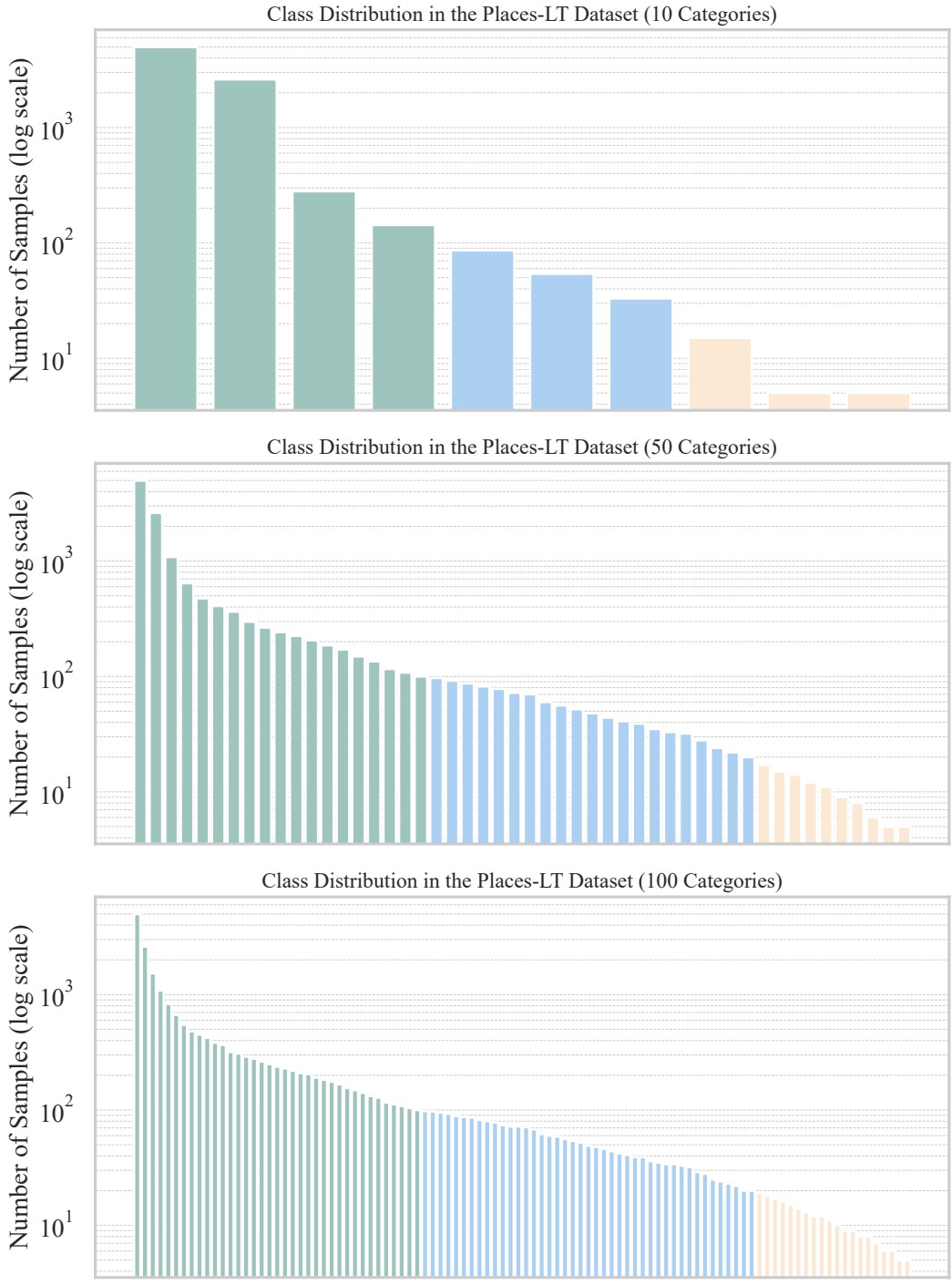

Figure 5: The Sample Distribution Characteristics of Various Datasets Generated via Places-LT (IR 996) with 10, 50, and 100 Categories.

a very high imbalance ratio (996). More critically, our method demonstrates superior performance on the challenging Tail classes in all scenarios. For the 10-category dataset, Ours achieves a Tail accuracy of **62.0%**, significantly outperforming the next best method, LDAM-DRW (60.33%). As the number of classes increases to 50 and 100, while maintaining the extreme IR, the Tail task becomes even more challenging, and many baselines exhibit severely degraded performance (e.g., CE, CE-DRW, KPS, LORT showing near zero or very low tail accuracy). In these more complex pathological settings

Table 5: Top-1 accuracy on Custom Places-LT Datasets (IR 996) with 10, 50, and 100 Categories. The grayed-out column indicates the overall accuracy. **Bold** indicates the best performance while underline indicates the second best.

| Method | IR 996 / 10 Categories | | | | IR 996 / 50 Categories | | | | IR 996 / 100 Categories | | | |
|---|---|---|---|---|---|---|---|---|---|---|---|---|
| | Head | Medium | Tail | All | Head | Medium | Tail | All | Head | Medium | Tail | All |
| CE | 61.50 | 0.00 | 0.00 | 24.60 | 40.28 | 0.00 | 0.00 | 14.50 | 28.14 | 0.00 | 0.00 | 10.13 |
| BS | 93.25 | 56.00 | 39.00 | 65.80 | 66.17 | 51.00 | 35.60 | 53.38 | 53.39 | 43.27 | 32.95 | 44.85 |
| BCL | 89.00 | 82.00 | 56.00 | 77.00 | 77.22 | 62.23 | 39.10 | 63.00 | 63.78 | 58.52 | 50.70 | 58.85 |
| CE-DRW | 94.50 | 62.67 | 37.67 | 67.90 | 66.28 | 44.36 | 15.10 | 46.44 | 53.97 | 39.77 | 16.90 | 40.31 |
| LDAM-DRW | 91.00 | 77.00 | 60.33 | 77.60 | 76.83 | 62.73 | 43.10 | 63.88 | 62.75 | 55.75 | 44.30 | 55.97 |
| KPS | 91.00 | 85.67 | 15.67 | 66.80 | 77.28 | 48.77 | 3.90 | 50.06 | 63.56 | 45.77 | 8.90 | 44.80 |
| LORT | 95.75 | 14.00 | 0.00 | 42.50 | 50.61 | 0.32 | 0.00 | 18.36 | 36.11 | 3.70 | 0.05 | 14.64 |
| Ours | 85.00 | 77.50 | **62.0** | **83.60** | 60.22 | **76.71** | **58.50** | **67.13** | **47.92** | **59.81** | **53.55** | **54.27** |

(50 and 100 categories), Ours continues to achieve the highest Tail accuracy (**58.50%** and **53.55%** respectively), substantially leading the second best methods (LDAM-DRW at 43.10% and 44.30%).

Furthermore, Ours also shows strong performance on Medium classes (best on 50 and 100 categories) and competitive, albeit not always leading, performance on Head classes (best on 100 categories), demonstrating its ability to balance learning across the entire frequency spectrum. The strong overall performance (**83.60%**, **67.13%**, **54.27%**) underscores this capability.

These experiments on custom Places-LT datasets with varied, but modest, class counts further validate the effectiveness of our proposed framework in handling pathological long-tailed distributions beyond the specific domain of ZincFluor. They highlight our method's unique ability to maintain high accuracy on scarce tail classes while ensuring robust overall performance, a critical requirement for scientific discovery applications.

## C   Detailed Theoretical Analysis

This section provides a detailed theoretical analysis of our proposed framework, emphasizing its formulation as an implicitly constrained multi-objective optimization problem and the role of Smooth Objective Regularization (SOR) in achieving a dynamically balanced solution for pathological long-tailed recognition. Our approach deviates from methods solely focused on manipulating classification logits or sample/loss weights based on class frequencies, by instead directly influencing the optimization trajectory to balance competing objectives via a principled penalty mechanism.

We define the model parameters as $\theta \in \mathbb{R}^P$. The learning process is driven by three constituent loss functions defined on the dataset $\mathcal{D}$:

- $\mathcal{L}_1(\theta) = \mathcal{L}_{\text{CE,orig}}(\theta)$: Cross-Entropy loss on original data.
- $\mathcal{L}_2(\theta) = \mathcal{L}_{\text{CE,aug}}(\theta)$: Cross-Entropy loss on augmented data.
- $\mathcal{L}_3(\theta) = \mathcal{L}_{\text{B-SC}}(\theta)$: Balanced Supervised Contrastive Learning loss, which incorporates frequency-aware weighting $w_y$ for tail classes (details in Appendix Section **??**).

Let $\mathbf{L}(\theta) = [\mathcal{L}_1(\theta), \mathcal{L}_2(\theta), \mathcal{L}_3(\theta)]^T \in \mathbb{R}^3$ be the vector of constituent losses.

In the context of pathological long-tailed data, minimizing the simple sum $\sum_{k=1}^{3} \mathcal{L}_k(\theta)$ can lead to suboptimal solutions where one loss is significantly higher than others. Our framework implicitly aims to solve a constrained multi-objective optimization problem: finding $\theta^*$ that minimizes a primary objective while keeping all constituent losses below certain thresholds $\epsilon_k$. This is conceptually similar to:

$$\min_{\theta} \sum_{k=1}^{3} \mathcal{L}_k(\theta) \quad \text{s.t.} \quad \mathcal{L}_k(\theta) \leq \epsilon_k, \quad k = 1, 2, 3.$$

Directly solving this is challenging. We approximate this via a penalty method, augmenting the sum of losses with a term that penalizes the maximum of the constituent losses.

**Proposition 2 (LogSumExp as a Smooth Maximum)** *The LogSumExp (LSE) function,* $\text{LSE}(\mathbf{v}) = \log \sum_{i=1}^{M} \exp(v_i)$*, is a differentiable, convex approximation of the maximum function, satisfying* $\max_i v_i \leq \text{LSE}(\mathbf{v}) \leq \max_i v_i + \log M$ *for any vector* $\mathbf{v} = [v_1, \ldots, v_M]^T \in \mathbb{R}^M$*.*

Let $v_{\max} = \max_{i \in \{1,\ldots,M\}} v_i$. For the lower bound:

$$\sum_{i=1}^{M} \exp(v_i) \geq \exp(v_{\max}). \tag{9}$$

Taking the logarithm (monotonic):

$$\log\left(\sum_{i=1}^{M} \exp(v_i)\right) \geq \log(\exp(v_{\max})) = v_{\max}. \tag{10}$$

Thus, $\max_i v_i \leq \text{LSE}(\mathbf{v})$.

For the upper bound: Since $v_i \leq v_{\max}$ for all $i$, $\exp(v_i) \leq \exp(v_{\max})$. Summing:

$$\sum_{i=1}^{M} \exp(v_i) \leq \sum_{i=1}^{M} \exp(v_{\max}) = M \exp(v_{\max}). \tag{11}$$

Taking the logarithm:

$$\log\left(\sum_{i=1}^{M} \exp(v_i)\right) \leq \log(M \exp(v_{\max})) = \log M + \log(\exp(v_{\max})) = \log M + v_{\max}. \tag{12}$$

Thus, $\text{LSE}(\mathbf{v}) \leq \max_i v_i + \log M$. Differentiability and convexity are standard properties of LSE.

We define the Smooth Objective Regularization (SOR) term using the LSE function applied to our vector of constituent losses $\mathbf{L}(\theta)$:

$$\mathcal{L}_{\text{SOR}}(\theta) = \lambda_{\text{SOR}} \cdot \text{LSE}(\mathbf{L}(\theta)/\tau_{\text{SOR}}) \tag{13}$$

where $\lambda_{\text{SOR}} > 0$ and $\tau_{\text{SOR}} > 0$. In our implementation, we set $\tau_{\text{SOR}} = 1$ and $M = 3$, yielding:

$$\mathcal{L}_{\text{SOR}}(\theta) = \lambda_{\text{SOR}} \cdot \log\left(\exp(\mathcal{L}_1(\theta)) + \exp(\mathcal{L}_2(\theta)) + \exp(\mathcal{L}_3(\theta))\right). \tag{14}$$

Minimizing $\mathcal{L}_{\text{SOR}}$ directly penalizes the largest constituent loss, pushing it down relative to the others.

Our total training objective is defined as the sum of the constituent losses augmented by the SOR term:

$$\mathcal{L}_{\text{total}}(\theta) = \sum_{k=1}^{3} \mathcal{L}_k(\theta) + \mathcal{L}_{\text{SOR}}(\theta) \tag{15}$$

Substituting the expression for $\mathcal{L}_{\text{SOR}}$:

$$\mathcal{L}_{\text{total}}(\theta) = \sum_{k=1}^{3} \mathcal{L}_k(\theta) + \lambda_{\text{SOR}} \cdot \log\left(\sum_{j=1}^{3} \exp(\mathcal{L}_j(\theta))\right). \tag{16}$$

Minimizing $\mathcal{L}_{\text{total}}$ serves as a penalty method approximation to the constrained multi-objective problem. It drives down the sum of losses while simultaneously using $\mathcal{L}_{\text{SOR}}$ to keep the maximum loss value in check, acting as a soft "tail penalty" (when $\mathcal{L}_3$ is high) and a "smooth constraint" promoting balance across all objectives.

The dynamic balancing property is evident from the gradient of $\mathcal{L}_{\text{total}}(\theta)$:

$$\nabla_\theta \mathcal{L}_{\text{total}}(\theta) = \sum_{k=1}^{3} \nabla_\theta \mathcal{L}_k(\theta) + \nabla_\theta \mathcal{L}_{\text{SOR}}(\theta). \tag{17}$$

The gradient of the SOR term is derived using the chain rule. Let $L_k = \mathcal{L}_k(\theta)$:

$$\nabla_\theta \mathcal{L}_{\text{SOR}}(\theta) = \lambda_{\text{SOR}} \cdot \nabla_\theta \log\left(\sum_{j=1}^{3} \exp(L_j)\right)$$

$$= \lambda_{\text{SOR}} \sum_{k=1}^{3} \frac{\partial \log(\sum_{j=1}^{3} \exp(L_j))}{\partial L_k} \nabla_\theta L_k$$

$$= \lambda_{\text{SOR}} \sum_{k=1}^{3} \frac{\exp(L_k)}{\sum_{j=1}^{3} \exp(L_j)} \nabla_\theta \mathcal{L}_k(\theta).$$

Let $p_k(\theta) = \frac{\exp(\mathcal{L}_k(\theta))}{\sum_{j=1}^{3} \exp(\mathcal{L}_j(\theta))}$. These $p_k$ values form a probability distribution over the constituent losses, where $p_k$ is high when $\mathcal{L}_k$ is large. The total gradient becomes:

$$\nabla_\theta \mathcal{L}_{\text{total}}(\theta) = \sum_{k=1}^{3} \nabla_\theta \mathcal{L}_k(\theta) + \lambda_{\text{SOR}} \sum_{k=1}^{3} p_k(\theta) \nabla_\theta \mathcal{L}_k(\theta). \tag{18}$$

Rearranging terms:

$$\nabla_\theta \mathcal{L}_{\text{total}}(\theta) = \sum_{k=1}^{3} \left(1 + \lambda_{\text{SOR}} p_k(\theta)\right) \nabla_\theta \mathcal{L}_k(\theta). \tag{19}$$

Equation 19 reveals the dynamic balancing. The gradient of each constituent loss $\nabla_\theta \mathcal{L}_k$ contributes to the total gradient with a weight $(1 + \lambda_{\text{SOR}} p_k)$. When a specific loss $\mathcal{L}_k$ becomes significantly larger than others, $p_k \to 1$, and the weight $(1 + \lambda_{\text{SOR}} p_k) \to 1 + \lambda_{\text{SOR}}$, effectively amplifying the gradient $\nabla_\theta \mathcal{L}_k$. Conversely, for a small loss $\mathcal{L}_j$, $p_j \to 0$, and its gradient is weighted by approximately 1. This mechanism ensures that the optimization actively targets the largest loss component, pulling it down.

This adaptive weighting, governed by the relative magnitudes of $\mathcal{L}_1, \mathcal{L}_2, \mathcal{L}_3$, imposes the "smooth constraint" by discouraging any single loss from dominating. For pathological long-tails, this is crucial: it prevents the model from solely optimizing the easily satisfied CPO on head classes while neglecting the critical $\mathcal{L}_3$ for tail classes, and vice-versa. Instead, it promotes a balanced decrease across all objectives, leading to a more robust model capable of deciphering the challenging extremes. This principled approach, rooted in approximating constrained multi-objective optimization via SOR's gradient dynamics, provides a theoretical basis for our method's superior performance on pathological long-tailed data.

