# OpenReview forum: "Deciphering the Extremes: A Novel Approach for Pathological Long-tailed Recognition in Scientific Discovery"
_NeurIPS.cc/2025/Conference — NeurIPS 2025 spotlight_

### Official Review · Reviewer_7mHX · 2025-06-28

**Clarity:** 2
**Significance:** 3
**Originality:** 2
**Rating:** 5
**Confidence:** 4

**Summary:**

This paper addresses a long-tail distribution problem by proposing a novel framework that combines Balanced Supervised Contrastive Learning and Smooth Objective Regularization, formulated as a multi-objective optimization problem. The key point of the proposed method is the use of the log sum exponential loss function to handle a multi-objective optimization problem. Experiments on both real-world and synthetic datasets show significant improvements compared to prior methods.

**Questions:**

Q1. What would be the performance difference if LogSumExp in SOR were replaced by a simple weighted sum?

Q2. Why were widely adopted long-tailed benchmarks like ImageNet-LT and iNaturalist not used? Can the proposed method generalize to more standard LTR scenarios?

Q3. Why were strong recent baselines like RIDE, MiSLAS, and SADE omitted from the comparison?

Q4. How were the key hyperparameters tuned, and how sensitive is the model performance to them? Without sensitivity analysis, it is difficult to assess whether the proposed method (with numerous hyperparameters) can be applied to new datasets in real-world scenarios.

Q5. Given the focus on tail-class recognition, why does the paper not include detailed error analysis such as tail-class confusion matrices or Top-K misclassifications?

**Ethical Concerns:**

["NO or VERY MINOR ethics concerns only"]

**Final Justification:**

The concerns regarding generalization, comparisons with additional baselines, and the ablation study on LogSumExp have been satisfactorily addressed.

**Limitations:**

yes

**Paper Formatting Concerns:**

no concerns

**Quality:**

3

**Strengths And Weaknesses:**

Strenghts:

(+): The use of LogSumExp in SOR provides a reasonable method to yield balanced states across multiple objectives, beyond simple loss aggregation.

(+): The proposed method achieves strong results on a real-world chemical dataset, with appropriate ablation experiments to validate the framework’s components.

Weakness:

(-): The study lacks evaluations on standard long-tailed benchmarks such as ImageNet-LT and iNaturalist, which are critical for demonstrating generalization and scalability.

(-): Key recent baselines in long-tail distributions like RIDE, MiSLAS, and SADE are missing from the comparisons, despite their strong performance and structural similarity to the proposed method.

(-): Although the paper emphasizes the role of LogSumExp, it does not include an ablation study comparing LogSumExp to a simpler weighted sum formulation of loss components, leaving the benefits of LogSumExp unverified.

(-): The paper introduces many additional important hyperparameters, but offers no details or sensitivity analysis, raising questions about reproducibility and robustness.

---

> ### Author Rebuttal · Authors · 2025-07-31
>
> ### **Response to Weaknesses and Questions**
>
> #### **Q1. Difference in performance if LogSumExp and a simple weighted sum.**
>
> This is a very critical question. The difference is not only reflected at the theoretical level but also manifests in the optimization process and final performance.
>
> 1. Theoretical
>
> **Simple Weighted Sum**: A method like $L_{total} = w_1 L_{CE} + w_2 L_{B-SC}$ represents a **static, pre-defined trade-off**. The weights $w_1$ and $w_2$ must be determined before training. However, in the pathological long-tail scenarios of unknown scientific tasks, a simple weighted sum can easily get stuck in a suboptimal solution due to improper weight settings.
>
> **Our SOR Method (LogSumExp)**: In contrast, our SOR term stems from the **Min-Max idea of multi-objective optimization**, aiming to minimize the worst-performing objective. The core advantage of LogSumExp, as a smooth approximation of the max function, lies in its **dynamic balancing**. If any loss term (e.g., $L_{B-SC}$ surging due to overfitting) becomes excessively large, the gradient of the SOR term will predominantly act on this "out-of-control" term, automatically and dynamically pulling it back to a reasonable level.
>
> 2. Optimization
>
> A simple weighted sum faces a major practical challenge: the **complexity of hyperparameter search**. For N loss terms, N-1 weight parameters need to be tuned. These weights interact with each other, making it nearly impossible to find a "universal" weight combination that performs well across all datasets and imbalance ratios. This greatly reduces the method's **practicality and reproducibility**. The entire SOR term introduces only **one key hyperparameter, $\lambda_{SOR}$**. This single parameter controls the overall strength of the "dynamic balancing system" without having to deal with the complex relative weights between individual loss terms.
>
> 3. Experimental Analysis
>
> We conducted a series of comparative experiments on CIFAR-10-LT, comparing our full method with a "simple weighted sum" version (where we set the weight of $L_{B-SC}$ to 1, a common baseline setting) and other SOTA methods.
>
> #### **Performance Comparison on CIFAR-10-LT (All: Overall Accuracy, Tail: Tail Class Accuracy)**
>
> | **Method** | **IR=100** | | | | **IR=200** | | | | **IR=500** | | | | **IR=1000** | | | |
> | :--- | :---: | :---: | :---: | :---: | :---: | :---: | :---: | :---: | :---: | :---: | :---: | :---: | :---: | :---: | :---: | :---: |
> | | Head | Medium | Tail | All | Head | Medium | Tail | All | Head | Medium | Tail | All | Head | Medium | Tail | All |
> | **Simple Weighted Sum** | 86.32 | 82.25 | - | 85.51 | 82.07 | 60.10 | - | 75.48 | 83.62 | 72.75 | 28.35 | 70.39 | 82.26 | 68.55 | 2.50 | 63.57 |
> | **Ours** | 87.59 | 89.90 | - | **88.04** | 84.05 | 84.33 | - | **84.14** | 81.68 | 79.64 | **59.39** | **77.94** | 76.80 | 76.60 | **38.99** | **69.20** |
> | *SOTA (SADE)* | 87.14 | 85.05 | - | 86.72 | 84.74 | 68.23 | - | 79.79 | 84.33 | 72.20 | 28.60 | 70.76 | 83.80 | 61.90 | 5.60 | 63.78 |
>
> The table clearly shows that as the imbalance ratio (IR) increases, the advantage of our method over the "simple weighted sum" becomes **increasingly huge**:
> *   At **moderate imbalance (IR=500)**, our method's tail accuracy (**59.39%**) is **more than double** that of the simple weighted sum (28.35%).
> *   At **extreme imbalance (IR=1000)**, the difference is even more staggering. The simple weighted sum almost completely fails on tail classes, with an accuracy of only 2.50%, and may require re-tuning to adapt. This robustness in extreme scenarios is one proof of our dynamic balancing strategy's effectiveness.
>
> ---
>
> #### **Q2. Why not use widely adopted long-tail benchmarks like ImageNet-LT and iNaturalist?**
>
> Thank you for this important question.
>
> First, we want to reiterate the **core motivation and specific focus** of our work. As emphasized in the paper, our research is **explicitly and intentionally** targeted at a specific, under-studied problem prevalent in scientific discovery: **"pathological long-tailed recognition."** We define this as scenarios characterized by: (1) **extreme class imbalance** (e.g., IR > 500); (2) a **limited number of classes** (e.g., < 20); and (3) a **limited overall sample size**. This setup is designed to simulate the real-world challenge of finding "needles in a haystack" in fields like materials science and rare disease diagnostics.
>
> Therefore, our initial experimental design (focusing on ZincFluor and extremely imbalanced CIFAR-LT variants) was intended to deeply investigate and solve this specific yet crucial "pathological" problem. "General" long-tail benchmarks like ImageNet-LT and iNaturalist, with their large scale, many classes, and large data volumes, do not fully represent this "data-scarce and extremely imbalanced" pathological condition we initially aimed to address.
>
> Nevertheless, we recognize the importance of your question regarding whether our method can generalize to more standard LTR scenarios, as this helps test its competitiveness in a broader context. During the rebuttal period, we have made our best effort to **supplement our experiments on the iNaturalist 2018 dataset**. The results are as follows:
>
> ##### **Performance Comparison on iNaturalist 2018**
>
> | Method | Top-1 Accuracy (%) |
> | :--- | :---: |
> | *MiSLAS [ICCV'21] | 70.7 |
> | *RIDE [ICLR'21] | 71.8 |
> | *SADE [CVPR'22] | 72.9 |
> | **Ours** | **74.9** |
> \* Performance data marked with an asterisk is cited from "Self-Supervised Aggregation of Diverse Experts for Test-Agnostic Long-Tailed Recognition" [CVPR'23].
>
> The core mechanism of our method—dynamically balancing competing learning objectives via **Smooth Objective Regularization (SOR)**—is also applicable in more general long-tail scenarios. We greatly appreciate this suggestion. We will include this new experimental result and its analysis in the appendix of the final version to fully demonstrate the robustness and broad applicability of our proposed framework.
>
> ---
>
> #### **Q3. Why were powerful recent baselines like RIDE, MiSLAS, and SADE omitted from the comparison?**
>
> Regarding the choice of baselines, in our initial experimental design, because our work **explicitly focuses on the specific scope of "pathological long-tailed recognition"** (i.e., extreme imbalance, few classes, limited sample size), we primarily chose representative LTR methods with significant conceptual differences as baselines to study the effectiveness of traditional strategies. Nevertheless, we completely understand the necessity of comparing our method with these strong recent baselines to more comprehensively demonstrate its competitiveness.
>
> We have made our **best effort to supplement direct comparisons with RIDE, MiSLAS, and SADE on the more challenging CIFAR-100-LT benchmark**. All experiments were run under our unified, standardized settings to ensure a fair comparison.
>
> ##### **Performance Comparison on CIFAR-100-LT (All: Overall Accuracy)**
>
> | **Method** | **IR=100** | **IR=200** | **IR=500** |
> | :--- | :---: | :---: | :---: |
> | **Simple Weighted Sum** | 43.25 | 44.45 | 39.14 |
> | **Ride** | 42.71 | 40.74 | 36.72 |
> | **MiSLAS** | 45.39 | 42.53 | 40.69 |
> | **SADE** | 44.70 | 42.86 | 37.21 |
> | **Ours** | **56.37** | **51.02** | **43.37** |
>
> We will include this important set of comparative experiments in the final manuscript. Thank you again for your valuable suggestion.
>
> ---
>
> #### **Q4. How are the key hyperparameters tuned, and how sensitive is the model's performance to them?**
>
> Our method, through its theoretical design, **simplifies the complexity of hyperparameter tuning**. We do not use the traditional "simple weighted sum" strategy, which requires manually tuning a large number of relative weights (like $w_1, w_2, ...$) for multiple loss terms. Instead, we **formalize the problem as a constrained multi-objective optimization problem** (as in Eq. 3 of the paper) and then transform it into an easy-to-solve unconstrained problem by introducing a **Smooth Objective Regularization (SOR) term based on LogSumExp**. Ultimately, our total loss function contains **only one key, tunable hyperparameter, $\lambda_{SOR}$**, which controls the strength of the entire "dynamic balancing system." This simplifies a complex multi-dimensional weight search problem into a single-dimensional strength adjustment problem.
>
> In our response to Q1, we have already demonstrated through experiments the effectiveness of our design. Due to time constraints, we were unable to provide an exhaustive sensitivity analysis during the rebuttal period. We are committed to supplementing the final manuscript with a **sensitivity analysis** of the key hyperparameter $\lambda_{SOR}$ to fully demonstrate the robustness and ease of application of our method in real-world scenarios.
>
> ---
>
> #### **Q5. Given the focus on tail-class recognition, why does the paper not include a detailed error analysis?**
>
> Thank you for this valuable suggestion. We have **preliminarily supplemented a confusion matrix analysis for the two rarest tail classes in the CIFAR-10-LT (IR=1000) setting**. The analysis shows that our method can successfully identify a considerable number of tail samples, with the main confusion occurring between the two equally rare classes.
>
> ##### **Tail Class Confusion Matrix**
> When CIFAR-10 IR=1000, we have 2 tail classes, and we achieve an accuracy of 38.99% on these classes. Below is our confusion matrix for the tail classes.
> $$
> \begin{bmatrix}
> 478 & 748 \\\\
> 473 & 301 \\\\
> \end{bmatrix}
> $$
>
> This preliminary analysis has already provided us with valuable insights. We will include this in the final manuscript and are committed to conducting more detailed error analyses (e.g., Top-K misclassifications) to more comprehensively demonstrate the behavior and potential areas for improvement of our method.

---

> > ### Comment · Reviewer_7mHX · 2025-08-02
> > **Response to authors**
> >
> > Thanks to the authors for their efforts in addressing the comments and questions. I truly appreciate the thorough revisions, which have successfully resolved most of my concerns. Accordingly, I have decided to raise my score.

---

### Official Review · Reviewer_GXBa · 2025-06-30

**Clarity:** 3
**Significance:** 3
**Originality:** 3
**Rating:** 5
**Confidence:** 4

**Summary:**

This paper focus on the challenge of pathological long-tailed recognitions, where rare but scientifically important instances are often underrepresented.
The authors propose a framework that combines Balanced Supervised Contrastive Learning (B-SCL) with Smooth Objective Regularization (SOR) to improve tail-class representation.
Experiments on the real-world ZincFluor dataset and synthetic CIFAR-LT variants show improved performance, especially in tail-class accuracy.

**Questions:**

1. Line 150, is Target 1 referring to Eq. 3?

2. The quantitative results show notable drops in performance on head classes. Could the authors provide a deeper analysis of this trade-off?

3. The composition of Places-LT does not follow the full 365-class structure of the original Places365. Does reducing the number of categories lower the difficulty of the dataset?

4. In Eq. 3, are $\epsilon_1$ $\epsilon_2$ and $\epsilon_3$  pre-defined hyperparameters? Or trainable parameters?

**Ethical Concerns:**

["NO or VERY MINOR ethics concerns only"]

**Final Justification:**

After reading the rebuttal from the authors and the comments from other reviewers, I believe the authors have addressed the concerns well. I acknowledge their efforts and will raise my score to Accept.

I recommend the authors incorporate the following revisions in the final version:​​

- ​Clarify ambiguous descriptions, particularly regarding W1 and Q1;
- ​Including Q4 response.

**Limitations:**

See Question 2. Additionally, the proposed method involves several hyperparameters, which may affect its stability and generalizability.

**Quality:**

3

**Strengths And Weaknesses:**

strengths:
1. The paper introduces a new real-world benchmark, the ZincFluor dataset, which captures the severe imbalance characteristics common in scientific discovery scenarios. This contribution can be valuable to the community.

2. The formal description for the identification of pathological long tails appears to be principled and well-structured.

3. The combination of different strategies is targeted and has a clear design motivation.


weakness:

1. The definition of $\mathcal{T}$ (E1.8) is ambiguous and possibly misleading.
For a fixed imbalance ratio, increasing the number of classes reduces $\mathcal{T}$.
Does a lower $\mathcal{T}$ imply a less pathological distribution, or a harder dataset due to more classes with fewer samples each? If T cannot be reliably interpreted as a measure of dataset difficulty, its utility is unclear.

2. Although the author(s) use LogSumExp, they do not specify whether they employ a numerically stable implementation.
Providing more discussion, such as on implementation details or numerical stability, will be helpful.


3. The real-world evaluation is limited to a single dataset (ZincFluor), and the synthetic benchmarks are relatively small-scale (CIFAR-10/100-LT).

---

> ### Author Rebuttal · Authors · 2025-07-31
>
> ### **Response to Reviewer GXBa**
>
> We sincerely thank Reviewer GXBa for the meticulous review and the valuable questions raised. Your insights have helped us identify and clarify key points in our paper. Below are our responses to the weaknesses and questions you have raised.
>
> *   **W1: On the ambiguity of the definition of $\mathcal{T}$ (Eq. 8)**
>
>     We completely agree with your observation. The **intention** behind our custom metric $\mathcal{T}$ was to capture a specific aspect of the "pathological long-tail" we define: that extreme class imbalance (reflected by $N_{majority}/N_{minority}$) is **concentrated within a few classes**. The $N_{classes}$ term in the denominator was meant to "penalize" datasets with many classes, thereby highlighting the "few classes, extreme imbalance" scenario we focus on, making the T-value higher in this specific context. This stems from an implicit bias we identified in our practical experience with special requirements in the field of science, where, due to the small total number of classes, the sample size disparity among the few classes is more pronounced, a challenge that existing strategies struggle to address.
>
>     We acknowledge that in more general long-tail scenarios, this metric may not fully or reliably explain the dataset's difficulty. **Thank you very much for your suggestion. In the final version, we will more rigorously explain the specific scope and limitations of this metric, as well as its distinction from the imbalance ratio, to avoid confusion.**
>
> *   **W2: On the numerical stability of LogSumExp**
>
>     You've raised a very important implementation detail. Yes, we did consider using a scaling factor to prevent numerical explosion. However, in our experiments, given the framework and the range of loss values, we found that numerical overflow situations hardly ever occurred. Therefore, for the sake of simplicity, we did not include this extra hyperparameter in the final version. **We will discuss this point further in the appendix of the subsequent manuscript to ensure the full reproducibility of our work.**
>
> *   **W3: Evaluation is limited to a single real-world dataset and smaller synthetic benchmarks**
>
>     We understand this limitation. To demonstrate the generalization capability of our method and its competitiveness on larger-scale benchmarks, we have **made our best effort to supplement experiments on the widely-adopted iNaturalist 2018 dataset**.
>
> ##### **Performance Comparison on iNaturalist 2018**
>
> | Method | Top-1 Accuracy (%) |
> | :--- | :---: |
> | *MiSLAS [ICCV'21] | 70.7 |
> | *RIDE [ICLR'21] | 71.8 |
> | *SADE [CVPR'22] | 72.9 |
> | **Ours** | **74.9** |
> \* Performance data marked with an asterisk is cited from "Self-Supervised Aggregation of Diverse Experts for Test-Agnostic Long-Tailed Recognition" [CVPR'23].The results show that our method also achieves state-of-the-art performance on this large-scale benchmark, demonstrating its strong generalization ability.
>
> ### **Response to Questions**
>
> *   **Q1: In line 150, does objective 1 refer to Equation 3?**
>
>     **Yes, objective 1 conceptually corresponds to the main optimization objective in Equation 3.** We apologize for not making this clearer in the original text. We will revise the wording in the final version to explicitly state this correspondence.
>
> *   **Q2: A deeper analysis of the trade-off regarding the performance drop in head classes**
>
>     This is a very insightful observation. Essentially, our method can be viewed as a **constrained optimization problem that leans more towards learning hard samples (driven by the B-SCL term) while ensuring overall performance is maintained (constrained by the CPO term)**. This is fundamentally different from many existing strategies, which typically still prioritize maximizing overall performance and only add some balancing strategies as an auxiliary. Therefore, our method may appear to place less emphasis on head-class performance in some scenarios. This is a **principled trade-off** made to achieve significant gains in tail-class performance, and the dynamic constraint ensures the degree of this sacrifice is managed. This also aligns with the real-world requirements of the "pathological long-tail" we define (e.g., in scientific discovery), where the value of identifying rare but critical instances far outweighs gaining a few more points of accuracy on common instances.
>
> *   **Q3: The composition and difficulty of Places-LT**
>
>     Regarding Places-LT, reducing the number of classes does indeed change the dataset's characteristics. For the **imbalance ratio (IR)**, since the denominator (number of samples in the minority class) and the numerator (number of samples in the majority class) remain unchanged, the IR value itself is not affected by the reduction in the number of classes. However, for our custom metric $\mathcal{T}$, since the denominator includes $N_{classes}$, reducing the number of classes would increase the $\mathcal{T}$ value, which, from the "intent" of our metric, represents a more "pathological" problem. Of course, the overall difficulty of a dataset is a complex concept that depends not only on the degree of imbalance but also on the number of classes, inter-class similarity, and other factors. For a large dataset, the change in difficulty may not be solely attributable to this.
>
> *   **Q4: The nature of $\epsilon_1$, $\epsilon_2$, and $\epsilon_3$ in Equation 3**
>
>     **They are neither pre-defined hyperparameters nor trainable parameters.** Equation 3 is our way of **conceptually formalizing** the problem, representing our approach of converting the original problem into a constrained optimization problem. Here, $\epsilon_i$ represents an **implicit, ideal constraint boundary**. In the actual optimization process, because we use **LogSumExp as a smooth approximation to the Min-Max strategy**, we bypass the need to directly handle or set these $\epsilon_i$. These constraint boundaries are **automatically and implicitly managed** by the penalty mechanism of the SOR term, without us needing to know their specific values.

---

> > ### Comment · Reviewer_GXBa · 2025-08-02
> > **Increase the score to 5**
> >
> > I appreciate the authors' effort, particularly the supplementary experiments on iNaturalist 2018. The authors made a diligent effort that has addressed my concerns about the scope of evaluation. Additionally, the principled explanation of the head-tail performance trade-off and the clarification on the conceptual nature of the constraints have clearly resolved my questions. I am pleased to raise my score to 5.

---

### Official Review · Reviewer_FP2j · 2025-07-02

**Clarity:** 4
**Significance:** 3
**Originality:** 4
**Rating:** 5
**Confidence:** 5

**Summary:**

This work presents a meticulously designed and empirically robust framework, explicitly tailored to address the critical domain of pathological long-tailed recognition within scientific datasets.

**Questions:**

**Questions**

1. Could the authors elaborate on the practical considerations or potential challenges in applying this framework to scientific datasets where the 'tail' is not just rare but also inherently noisy or ambiguously labeled?
2. Given the multi-objective nature, are there specific criteria or metrics beyond accuracy that could be used to quantify the 'Pareto-optimality' achieved by the 'L\_total' objective, especially in a scientific discovery context where different types of errors might have varying costs?
3. How might the class-frequency aware re-weighting in B-SCL be adapted if the class frequencies are not precisely known a priori, which might be the case in some emerging scientific discovery scenarios?

**Ethical Concerns:**

["NO or VERY MINOR ethics concerns only"]

**Final Justification:**

This paper is clear to me and all concerns are addressed in the rebuttal.

**Limitations:**

Yes

**Quality:**

4

**Strengths And Weaknesses:**

**Strengths:**

1. The paper's strength lies in its elegant formalization of pathological LTR as a multi-objective optimization problem. This principled approach, and the subsequent derivation of the SOR term, provides a strong theoretical underpinning for balancing competing objectives.
2. The synergistic design of B-SCL and SOR is highly effective. B-SCL, with its class-frequency aware re-weighting, directly tackles the representation bottleneck for rare classes, while SOR acts as a dynamic arbiter, preventing overfitting to scarce data or dominance by head classes. The ablation studies convincingly validate the contribution of each component.
3. The introduction of ZincFluor, a real-world dataset exhibiting the 'pathological' characteristics, provides invaluable validation. The use of CIFAR-LT variants with tunable imbalance ratios allows for systematic benchmarking across varying degrees of severity, thoroughly demonstrating the method's robustness.

**Weaknesses:**

1. While the 'TSOR = 1' simplification is mentioned, a more detailed analytical or empirical sensitivity analysis of ASOR and TSOR to different dataset characteristics (e.g., number of classes, total sample size, degree of imbalance) would provide deeper insights into the framework's generalizability and practical tuning.
2. The paper utilizes a standard SupCon foundation. A brief discussion or comparison with more advanced contrastive learning techniques, or how they might be integrated into the B-SCL framework, could further highlight its extensibility and potential for future enhancements.

---

> ### Author Rebuttal · Authors · 2025-07-31
>
> ### **Response to Reviewer FP2j**
>
> We are very grateful to Reviewer FP2j for the deep understanding and high praise of our work. The questions you've raised are profound and directly address the core considerations of our method in practical applications. Below are our detailed responses to your points.
>
> *   **W1: Sensitivity analysis of $\lambda_{SOR}$ and $\tau_{SOR}$**
>
>     This is a very pertinent suggestion. Although we set $\tau_{SOR}=1$ for simplification in the paper, a more in-depth analysis of these two hyperparameters would indeed provide valuable insights.
>
>     **Theoretical Analysis:**
>     *   **$\tau_{SOR}$ (Temperature Coefficient)**: In the LogSumExp function, $\tau$ acts as a smoothness regulator. As $\tau \to 0$, the LSE function becomes "sharper," approaching the non-differentiable `max` function. As $\tau \to \infty$, the LSE becomes "flatter," approaching the average of all loss terms. Thus, $\tau$ controls the penalty strength of our SOR term on the "worst-performing" objective. A smaller $\tau$ implies a stricter "min-max" constraint, while a larger $\tau$ is closer to simple loss aggregation. Our choice of $\tau_{SOR}=1$ is a balanced one in both theory and practice, as it maintains focus on the worst objective while ensuring sufficient smoothness for optimization.
>     *   **$\lambda_{SOR}$ (Regularization Strength)**: This parameter controls the weight of the entire SOR penalty term. A larger $\lambda_{SOR}$ means the model will more aggressively balance the objectives to prevent any single one from becoming too large, while a smaller $\lambda_{SOR}$ implies less intervention, closer to the sum of the base objectives.
>
>     We will provide the relevant analysis for your review as soon as possible during the rebuttal period.
>
> *   **W2: Integration with more advanced contrastive learning techniques**
>
>     You've pointed out an excellent future direction. Our B-SCL framework is designed to be **modular and extensible**. It is entirely possible to integrate more advanced contrastive learning strategies to achieve stronger feature extraction capabilities, thereby better serving scientific problems. For example, when extending our method to fields like **genomics**, we could incorporate more complex Graph Neural Network (We have done this in ZincFluor to better adapt to the molecular field) architectures or self-attention mechanisms (like Transformers) as feature extractors to better capture key patterns from complex gene regulatory networks. We will discuss these potential extensions in the conclusion to highlight the future potential of our framework.
>
> *   **Q1: Practical considerations for applying the framework to noisy or ambiguously labeled scientific datasets**
>
>     This is a very practical challenge. In real-world scientific exploration, tail data is not only scarce but often accompanied by high noise or label uncertainty (e.g., the fluorescence intensity of a molecule might be mislabeled due to experimental fluctuations). Our framework has certain intrinsic advantages in this regard:
>     1.  **Regularization Effect of SOR**: Our **SOR mechanism, through dynamic balancing across different batches, prevents any single loss term (including $L_{B-SC}$, which is dominated by tail samples) from becoming excessively large**. This means that even if $L_{B-SC}$ attempts to surge due to noisy samples, the SOR term will impose a strong penalty, pulling it back and thus suppressing overfitting to noise.
>     2.  **Relative Learning of B-SCL**: Contrastive learning inherently learns the **relative relationships** between samples rather than fitting an absolute label. It focuses on "sample A is more similar to sample B than to sample C." This learning paradigm is more robust to isolated label noise compared to cross-entropy loss, which directly fits one-hot labels.
>
> *   **Q2: Specific metrics to quantify "Pareto optimality"**
>
>     In the multi-objective optimization problem of long-tail recognition, the "Pareto optimality" of a solution is manifested in finding the best balance between the performance on head classes and tail classes. If we wish to construct a metric different from accuracy, we might assign weights to different classes based on the specific task to better quantify the Pareto front.
>
>     Beyond accuracy, we can use more task-oriented metrics. For example, in **new material discovery**, misclassifying an unstable compound (typically a head/medium class) as "high-potential stable" (a rare tail class, a false positive) could lead to significant investment of R&D resources in the wrong direction, with a cost far greater than missing a truly potential material (a false negative). We could define a **cost-sensitive loss** that heavily penalizes false positive predictions for tail classes. In this case, quantifying "Pareto optimality" becomes a matter of finding the best balance between overall prediction accuracy and the **total expected R&D cost**.
>
> *   **Q3: The case where class frequencies are not precisely known a priori**
>
>     This scenario partially highlights the advantages of our framework. In situations with unknown or dynamically changing class frequencies, relying on a fixed re-weighting or re-sampling strategy would lead to very unstable performance. However, our **SOR dynamic balancing mechanism does not directly depend on precise class frequencies**. Its role is to monitor the magnitude of various learning objectives (like classification loss and contrastive loss). If an objective (e.g., the tail contrastive loss) suddenly increases due to unknown changes in the class distribution, SOR will **automatically and dynamically** apply a penalty to prevent the model from spiraling out of control. This adaptive balancing capability allows our framework to provide an effective exploration when facing unknown or dynamic class distributions.

---

> > ### Comment · Reviewer_FP2j · 2025-08-02
> >
> > I greatly appreciate the authors' thorough and thoughtful rebuttal. Your response has not only resolved my initial reservations but has also reinforced my confidence in the practical utility of your proposed framework. It is clear the problem and method have been meticulously considered. This solidifies my positive evaluation, and I look forward to seeing the promised revisions in the camera-ready version.

---

### Official Review · Reviewer_a3v3 · 2025-07-02

**Clarity:** 4
**Significance:** 3
**Originality:** 4
**Rating:** 5
**Confidence:** 5

**Summary:**

This paper presents a very practical solution to the extremely imbalanced classification problem in science. The proposed scheme significantly improves the performance on the real-world ZincFluor dataset and CIFAR-10/100-LT, especially in identifying rare classes under extreme long-tailed distributions.

**Questions:**

Please refer to the weaknesses proposed in strength and weaknesses section.

And I have some minor questions:

- The model shows substantial gains on tail classes even under IR=1000, which is rare in related work. Do the authors believe the method can extend to cross-modal or higher-dimensional data such as protein structure prediction or astronomical spectra?

**Ethical Concerns:**

["NO or VERY MINOR ethics concerns only"]

**Final Justification:**

After reading the authors’ response, my concerns have been resolved. I believe it is a strong paper that deserves to be published at NeurIPS. Therefore, I have maintained my positive evaluation.

**Limitations:**

Limitations are discussed in the Appendix.

**Quality:**

4

**Strengths And Weaknesses:**

Strengths：

1. I think this paper has good practical value for real-world science tasks. The authors are encouraged to explore deployment strategies, parameter sensitivity, and dynamic tail identification in future work to improve community adoption.
2. The proposed approach fits the characteristics of real-world scientific data: the limited number of classes, the small total sample size, and the critical nature of tail classes that are difficult to learn.
3. The ZincFluor dataset is compelling and the experiments on CIFAR are rigorous.
4. I think the objective design of this paper is balanced and well structured.

Weaknesses：

1. Some related works need to be discussed in more depth. This will help to more fully explain why their proposed method is particularly suitable for this problem scenario, especially when compared to other methods. For example, [1, 2, 3, 4].
2. Furthermore, given that the concept of multi-objective optimization appears to be an inspiration for this paper, it would be beneficial to provide a more thorough comparison with relevant works in this area, such as, [5, 6].

[1] Rethinking Classifier Re-Training in Long-Tailed Recognition: A Simple Logits Retargeting Approach

[2] Harnessing Hierarchical Label Distribution Variations in Test Agnostic Long-tail Recognition

[3] Key Point Sensitive Loss for Long-Tailed Visual Recognition

[4] AUCSeg: AUC-oriented Pixel-level Long-tail Semantic Segmentation

[5] Pareto Deep Long-Tailed Recognition: A Conflict-Averse Solution

[6] Two Fists, One Heart: Multi-objective Optimization based Strategy Fusion for Long-tailed Learning

---

> ### Author Rebuttal · Authors · 2025-07-31
>
> ### **Response to Reviewer a3v3**
>
> We are very grateful to Reviewer a3v3 for the high praise of our work and the valuable suggestions provided. We are pleased that you recognize the practical value of our method in solving real-world scientific tasks, as well as the rigor of our experimental design. Below are our detailed responses to the weaknesses you have raised.
>
> #### **Response to Weaknesses**
>
> *   **W1: On the deeper discussion with related works (Refs [1-4])**
>
>     Thank you very much for providing these high-quality related works. We agree that a deeper comparison with these works can better highlight the uniqueness of our method. These works [1-4] have indeed addressed important long-tail challenges in their respective domains, such as classifier retraining [1], test-time distribution shifts [2], specific loss designs [3], and semantic segmentation [4].
>
>     However, a key difference between our method and theirs lies in the **problem scenario** we focus on. Our work attempts to capture a specific aspect of the "pathological long-tail" we define: that extreme class imbalance is **concentrated within a few classes**. This scenario is common in scientific exploration, for example, when building surrogate models for molecular design, where the data is severely imbalanced, and the classes with the potential for new discoveries are extremely rare. This implies a bias in importance, where the value of identifying these extremely rare classes far outweighs that of others. Our proposed framework is designed to address this dilemma where "a few mainstream classes overwhelmingly dominate the entire dataset, while extremely valuable samples are exceptionally rare." At this level, the efforts of these other methods are not applicable, or rather, they aim to solve broad and general long-tailed distributions.
>
>     Despite the different focus, some of the excellent design ideas from these works, such as the logit adjustment strategy in [1] or the focus on key points in [3], could potentially be combined with our framework in the future to further enhance performance. **We will properly cite and discuss these works in the final version to more clearly define the scope and contribution of our method.**
>
> *   **W2: On the comparison with multi-objective optimization (MOO) related works (Refs [5,6])**
>
>     Thank you again for supplementing these key preceding works. Placing our method in the broader context of MOO research is very beneficial. These works do share similarities with ours, but the core objectives and levels are different:
>     *   **[5] Pareto-LTR** aims to use multi-objective optimization to resolve the intrinsic conflicts **between different classes**, especially the competitive relationship between head and tail classes during the learning process.
>     *   **[6] Two-in-One-Box** attempts, at a more abstract level, to pursue a Pareto-optimal performance upper bound by fusing different long-tailed learning strategies.
>
>     In contrast, the similarity with them lies in the introduction of multi-objective and trade-off ideas. However, our **multi-objective level and specific goals are not the same**. We do not directly address inter-class conflicts but rather treat **classification performance** and **tail representation learning** as two macroscopic objectives. Our core contribution is to use multi-objective ideas to **construct a novel optimization framework and a unique trade-off strategy** for the special scenario of "pathological long-tail" that we define.
>
> #### **Response to Questions**
>
> *   **Q1: Can the method be extended to cross-modal or higher-dimensional data?**
>
>     **We believe this is entirely possible.** The significant gains in tail-class performance at IR=1000 that you observed are a testament to the core strengths of our framework. The foundation of our framework (balanced contrastive learning for representation, multi-objective optimization for balance) is data-modality-agnostic, which gives it a broad scope of applicability and a unique perspective.
>
>     *   For **protein structure prediction**, we can encode the 3D structure or sequence information of proteins into graph or sequence data, and then use our framework to identify those structural motifs with rare but critical functions.
>     *   For **astronomical spectra**, we can treat them as one-dimensional signals and use our framework to find those extremely faint "tail" spectral features produced by unknown celestial objects or physical phenomena.
>
>     Admittedly, as Reviewer FP2j pointed out, when tackling these more challenging scientific problems, our core framework may need to be combined with **domain-specific components**, such as more powerful feature extractors designed for specific data types (e.g., GNNs or Transformers). We are very excited about this direction of expansion and believe our work provides a solid foundation for it.

---

> > ### Comment · Reviewer_a3v3 · 2025-08-02
> >
> > Thank you for your detailed and thoughtful responses. They effectively address my concerns and clarify the key points. I will maintain the positive evaluation.

---

### Note · Authors · 2025-08-13

Dear PCs, SACs, ACs, and Reviewers,

We would like to take this final opportunity to express our sincere gratitude for your insightful feedback and constructive discussions. This valuable process has significantly helped clarify our work's key aspects and greatly enhanced the completeness and depth of our paper.
Our work introduces a principled framework (B-SCL + SOR) specifically designed to tackle the critical challenge of "pathological long-tailed recognition" in scientific discovery—a scenario characterized by few classes, limited samples, and extreme imbalance.
Prompted by your valuable questions on generalization, we conducted extensive new experiments during the rebuttal period. We are pleased to report that our method achieves state-of-the-art performance on the large-scale iNaturalist 2018 benchmark and surpasses strong recent baselines like RIDE, MiSLAS, and SADE.

These new results demonstrate that our framework not only excels in the "pathological" scenarios it was designed for but also possesses strong competitiveness and generalizability in broader, standard LTR settings. We believe these additions have fully addressed the core concerns raised and significantly strengthen our contribution.

Thank you again for your time and guidance. We are committed to incorporating all discussed improvements and new results into the later  version.

---

### Decision · Program_Chairs · 2025-09-17

**Decision:**

Accept (spotlight)

**Comment:**

This paper tackles the challenge of pathological long-tailed distributions in scientific datasets. The authors propose a novel framework combining Balanced Supervised Contrastive Learning (B-SCL) and Smooth Objective Regularization (SOR) to enhance tail-class representation while balancing overall classification performance. Extensive experiments on both real and synthetic long-tailed benchmarks show significant improvements over existing methods, especially in identifying rare but important classes.

The paper has several strengths:

•	The paper introduces a novel and well-motivated framework specifically tailored to pathological long-tailed distributions in scientific data, going beyond generic LTR methods.

•	The experimental results are strong and convincing, with consistent improvements across real-world and synthetic benchmarks, especially on rare tail classes.

•	The methodological contributions are clear, with both B-SCL and SOR shown to be effective through ablation studies.

•	The paper is well-written and well-organized, and easy to follow.

After the rebuttal, most of the reviewers’ concerns have been satisfactorily addressed. The authors clarified theoretical aspects of the multi-objective formulation and provided more justification for the choice of contrastive learning components. They also conducted extensive experiments in various datasets and achieved SOTA performance during the rebuttal period. Reviewers generally agreed that these responses strengthened the paper and resolved their major reservations.

I recommend accept. The paper makes a significant and well-validated contribution to the problem of pathological long-tailed recognition in scientific discovery. That said, I strongly encourage the authors to incorporate the reviewers’ suggested revisions into the final version. Addressing these points will further strengthen the paper’s clarity, reproducibility, and long-term impact.